# DLM-One: Diffusion Language Models for One-Step Sequence Generation

## Abstract

This paper introduces *DLM-One*, a score-distillation-based framework for one-step sequence generation with continuous diffusion language models (DLMs). DLM-One eliminates the need for iterative refinement by aligning the scores of a student model's outputs in the continuous token embedding space with the score function of a pretrained teacher DLM. We investigate whether DLM-One can achieve substantial gains in sampling efficiency for language modeling. Through comprehensive experiments on DiffuSeq, a representative continuous DLM, we show that DLM-One achieves up to $\sim 500\times$ speedup in inference time while maintaining competitive performance on benchmark text generation tasks used to evaluate the teacher models. We further analyze the method's empirical behavior across multiple datasets, providing initial insights into its generality and practical applicability. Our findings position one-step diffusion as a promising direction for efficient, high-quality language generation and broader adoption of continuous diffusion models operating in embedding space for natural language processing.

## 1 Introduction

Recent progress in large language models (LLMs) has been primarily driven by autoregressive (AR) modeling, where sequences are generated token by token in a left-to-right fashion (Vaswani et al., 2017; Radford et al., 2018; Brown et al., 2020; Achiam et al., 2023; Chowdhery et al., 2022; Team et al., 2023; Touvron et al., 2023; Bai et al., 2023; Grattafiori et al., 2024). While AR models have demonstrated remarkable performance across a wide range of natural language processing (NLP) tasks, they suffer from several well-known limitations: exposure bias, error accumulation, lack of bidirectional context during generation, limited controllability in non-left-to-right scenarios, and inability to revise previously generated text (Keskar et al., 2019; Dathathri et al., 2020; Li et al., 2022a; Reid et al., 2022; Kaddour et al., 2023; Zhang et al., 2023; Bachmann & Nagarajan, 2024; Berglund et al., 2024). Moreover, certain data distributions may be inherently challenging to capture with AR models but can be modeled more effectively by alternative non-AR approaches, such as energy-based models (Lin et al., 2021). The sequential nature of token generation also imposes a fundamental bottleneck on inference speed, motivating the development of various acceleration techniques to reduce computational overhead (Khoshnoodi et al., 2024). These limitations have spurred growing interest in non-AR paradigms—particularly diffusion language models (DLMs)—which offer a fundamentally different approach by enabling parallel decoding of entire sequences instead of generating them one token at a time.

In contrast to AR LMs, which rely on causal attention and require one function evaluation (NFE) per token, DLMs often apply bidirectional attention and can generate sequences of predefined length in parallel (Li et al., 2022a; Strudel et al., 2022; Dieleman et al., 2022; Gong et al., 2022). Existing DLMs perform generation via iterative refinement, enabling all tokens in a sequence to interact with each other and allowing for holistic reasoning over the full sequence. The per-token computational cost of DLMs depends on both the NFEs used during the iterative refinement process and the length of the target sequence. By adjusting the sequence length during pretraining and the number of NFEs at inference time, DLMs offer flexible configurations to trade off generation quality and speed (Li et al., 2022a; He et al., 2023; Li et al., 2023b; Lin et al., 2023; Zheng et al., 2024b; Gao et al., 2024).

However, despite this flexibility, there is currently no conclusive evidence that DLMs can either generate faster while matching the performance of AR models, or achieve better performance at a comparable model size (Gulrajani & Hashimoto, 2024; Han et al., 2023; Mahabadi et al., 2024; Nie et al., 2025a;b; Gong et al., 2024). Nevertheless, there is substantial potential to accelerate DLMs

by significantly reducing the number of required NFEs—without sacrificing performance—through diffusion distillation techniques. Such techniques have recently shown notable success in speeding up continuous diffusion models for vision tasks (Sauer et al., 2024; Yin et al., 2024; Zhou et al., 2024b).

DLMs can be broadly categorized into two types: **discrete** and **continuous**. Discrete DLMs operate directly on categorical token spaces (Hoogeboom et al., 2021; Austin et al., 2021; He et al., 2023; Lou et al., 2024), aligning naturally with the symbolic nature of language. These models have demonstrated promising performance, *e.g.*, on unconditional text generation tasks. However, they still suffer from prohibitively slow sampling—often requiring hundreds to thousands of steps—due to the lack of effective acceleration techniques tailored to discrete diffusion. In contrast, this issue is less prominent in the vision domain, where continuous diffusion models and corresponding acceleration methods predominate.

Unlike discrete diffusion, continuous DLMs model the diffusion process in the embedding space, treating token representations as continuous vectors (Li et al., 2022a; Gong et al., 2022; Ye et al., 2023; Yuan et al., 2022; Gao et al., 2024; Gulrajani & Hashimoto, 2024). Their sampling process naturally supports controllability via auxiliary guidance (Dhariwal & Nichol, 2021; Ho & Salimans, 2022), and can be further accelerated while maintaining competitive performance (Song et al., 2021; Lu et al., 2022; Salimans & Ho, 2022). These properties make DLMs particularly appealing for real-world applications. Although they are arguably less aligned with the inherently discrete nature of language—which may explain their relatively limited adoption compared to discrete DLMs—they offer a key advantage: compatibility with a wide range of acceleration strategies developed in the vision domain, such as consistency distillation (Song et al., 2023; Song & Dhariwal, 2023; Geng et al., 2024) and score distillation (Poole et al., 2023; Wang et al., 2023; Luo et al., 2023; Yin et al., 2023; Zhou et al., 2024c). These methods enable one- or few-step generation with minimal quality degradation and, when enhanced with real data during distillation, can even surpass the teacher model (Zhou et al., 2025b).

This prompts a key question: ***Can similar substantial gains in sampling efficiency be realized in language generation?*** More specifically, can we generate a sequence of, *e.g.*, 100 tokens through a single forward pass of the diffusion backbone network? This would correspond to 100 NFEs for AR LMs, and potentially even more for existing DLMs, where the exact count depends on the number of iterative refinement steps but often reaches into the hundreds.

If so, it opens a promising research direction: how to pretrain stronger continuous DLMs that are naturally amenable to distillation. Potential approaches include improving the word embedding space or jointly optimizing it during pretraining. In this work, we focus on distilling existing continuous DLMs pretrained in the word embedding space, using publicly available checkpoints or open-source implementations, while leaving the design and pretraining of improved, larger models for future exploration. Specifically, we choose continuous DLMs pretrained with DiffuSeq (Gong et al., 2022) as our teacher models.

We consider continuous diffusion for language modeling and investigate whether vision-inspired distillation techniques can enable drastically more efficient, high-quality sequence generation. Specifically, we propose a score distillation-based framework for training *DLMs for one-step sequence generation* (DLM-One). Our method distills the knowledge of a pretrained teacher DLM into a student model of the same size that generates sequences in a single forward pass. Unlike prior work that often relies on hundreds of iterative refinement steps to produce a single sequence, DLM-One eliminates the need for iterative sampling altogether. It does so by aligning the scores of the student's outputs with the teacher's score function in the forward-diffused noisy space. To stabilize training and prevent degenerate solutions, we introduce an auxiliary adversarial loss and adopt a two-stage optimization scheme that progressively refines the student.

Under the same model size, DLM-One achieves up to $L\times$ speedup compared to AR LMs, where $L$ is the target sequence length. It also achieves up to $\text{NFEs}\times$ speedup over the teacher DLM, where NFEs denotes the number of iterative refinement steps used during teacher sampling. For example, in terms of wall-clock time, DLM-One delivers approximately $500\times$ speedup over DiffuSeq, while achieving comparable generation quality. These results redefine what is possible along the Pareto front between generation quality and sampling efficiency.

Our contributions are summarized as follows:

- We introduce **DLM-One**, a practical score distillation framework for continuous DLMs that enables one-step sequence generation without iterative denoising.

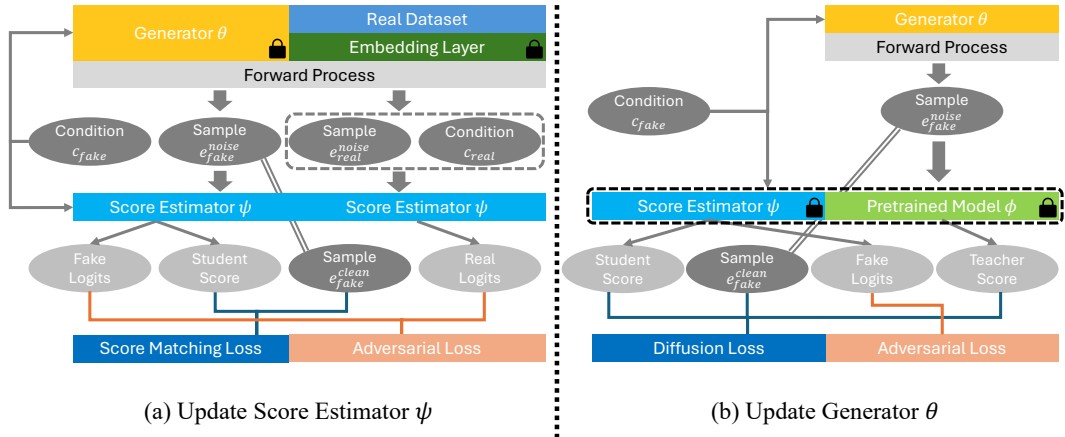

Figure 1: **Overview of the adversarial score distillation process. Left:** During score estimator $\psi$ updates, both real and generated data-condition pairs are used. The generator $\theta$ produces $e_{\text{fake}}^{\text{clean}}$ from $c_{\text{fake}}$, while real pairs are sampled from the dataset. The shared score estimator $\psi$ is trained for both score prediction and GAN discrimination. **Right:** During generator $\theta$ updates, the pretrained teacher model $\phi$ provides target scores, and $\psi$ produces both student scores and fake logits. These two scores are used to compute the score matching loss together with the clean data. Additionally, the generator is optimized to encourage the generation of more realistic samples under the feedback (i.e., logits) from $\psi$, via the adversarial loss. Modules marked with a 🔒 are frozen during the respective updates.

- We propose a two-stage training strategy with adversarial stabilization to enhance student quality and address common failure modes in distilling DLMs in a data-free setting.
- Our empirical evaluation on benchmark text generation tasks used by the teacher models demonstrates that our method achieves competitive performance while reducing sampling cost by up to $\sim500\times$ over DiffuSeq and up to $\sim256\times$ over Plaid.

## 2 RELATED WORK

### 2.1 DIFFUSION LANGUAGE MODELS

Unlike AR LMs, DLMs typically use a denoising score matching loss for training and predict entire sequences or multiple tokens at once. This eliminates the need for left-to-right, token-by-token sampling and enables faster decoding. Inspired by continuous diffusion models (Ho et al., 2020; Nichol & Dhariwal, 2021), Li et al. (2022b) propose an end-to-end language modeling approach that jointly learns word embeddings and a diffusion model in the embedding space, combining a diffusion loss with a rounding loss. Gong et al. (2022) adopt a similar strategy for sequence-to-sequence tasks by concatenating conditioning inputs with target sequences and modifying the forward diffusion process to apply noise only to the target. In contrast to the decoder-only architecture used in DiffuSeq, Yuan et al. (2022) introduce a dedicated encoder to process the conditioning input.

Viewing the additional rounding loss as a regularization term, Gao et al. (2024) propose an anchor loss to improve training stability and prevent embedding collapse. To bridge the likelihood gap, Gulrajani & Hashimoto (2024) introduce Plaid, the first DLM shown to achieve likelihood performance comparable to that of AR models on standard language modeling benchmarks. While these models all operate in the embedding space, we note that DLMs have also been trained in the vocabulary logit space (Han et al., 2023; Mahabadi et al., 2024) and the latent space of an encoder-decoder LM (Lovelace et al., 2023; Zhang et al., 2023; Zhou et al., 2024a; Shabalin et al., 2025). Extending DLM-One to such models represents a promising direction for future work.

In addition to continuous diffusion models, discrete diffusion models have also been studied for text generation. Hoogeboom et al. (2021) introduce a multinomial diffusion process for modeling categorical data. Austin et al. (2021) further explore various discrete state transition matrices, adding flexibility to the discrete diffusion process. By vector quantizing images into sequences of visual tokens (Oord et al., 2017; Esser et al., 2021), discrete diffusion models have been applied to generate visual token sequences that can be decoded back into images (Gu et al., 2022; Hu et al., 2022). Lou et al. (2024) extend score matching (Hyvärinen & Dayan, 2005) losses from

continuous to discrete spaces. Ou et al. (2025) reformulate the concrete score (Meng et al., 2022) as a product of time-independent conditional probabilities and a time-dependent scalar, enabling more efficient sampling. Rather than working on the general forward process, Sahoo et al. (2024) improve the practical performance of discrete DLMs by focusing on the masking strategy and introducing tight Rao-Blackwellized objectives. Shi et al. (2024) derive a simplified variational objective for continuous-time masked DLMs and generalize the masking schedule to support state dependency. Recognizing the connection between masked DLMs and AR models, Gong et al. (2024) propose a continual pretraining approach to adapt pretrained AR models into discrete DLMs. Nie et al. (2025b) introduces LLaDA that pretrains a large discrete DLM from scratch and further improves it with supervised fine-tuning.

## 2.2 FASTER DIFFUSION

Diffusion models are known for their strong generative capabilities; however, this comes at the cost of hundreds to thousands of NFEs during sampling in their original formulation (Ho et al., 2020; Song et al., 2020). Despite progress with training-free acceleration techniques, such as advanced samplers (Liu et al., 2022; Lu et al., 2022) and model quantization (Li et al., 2023a), diffusion models still lag behind traditional generative models like GANs and VAEs in terms of sampling speed.

Several directions have been explored to accelerate diffusion-based generation. Liu et al. (2024) and Guo et al. (2024) propose Discrete Copula Diffusion, which combines a discrete diffusion model with a copula-based correction module at inference time to improve the denoising distribution. Masked diffusion models (MDMs) (Zheng et al., 2024a) accelerate generation via a first-hitting sampling strategy. Progressive distillation (Salimans & Ho, 2022) introduces an iterative distillation scheme, reducing the number of sampling steps by progressively halving them. Luo et al. (2023) and Yin et al. (2024) propose minimizing the integral Kullback–Leibler divergence between the generative distributions of teacher and student models. From a score-distillation perspective, Zhou et al. (2024c) proposes a Fisher divergence-based distillation objective and an accompanying alternating optimization procedure that jointly enhance convergence and generation quality. Further improvements in data-free score distillation have been achieved by incorporating real data and adversarial training (Sauer et al., 2024; Yin et al., 2024; Zhou et al., 2025b).

In the context of accelerating DLMs, AR-Diffusion (Wu et al., 2023) incorporates autoregressive characteristics into diffusion models by allocating fewer refinement steps to earlier tokens, thereby better modeling sequential dependencies. Unlike training-free methods that focus on better utilizing the frozen teacher for faster inference, diffusion distillation trains a student model from a pretrained teacher, enabling generation in just one or a few inference steps. Our work—*DLM-One*—is a diffusion distillation framework that enables one-step sequence generation while preserving the generation quality of the teacher, effectively eliminating the need for iterative refinement.

## 3 ONE-STEP DIFFUSION LANGUAGE MODELS

To train a one-step sequence generation model, we begin with a pretrained teacher DLM that operates in a continuous embedding space. In this setting, each discrete language token is first mapped to a real-valued embedding vector via an embedding layer. The diffusion process is then applied to these continuous embeddings rather than to the discrete tokens themselves. This setup enables us to leverage well-established acceleration methods from continuous diffusion models in the vision domain, while focusing on language-model-specific adjustments essential for effective sequence generation.

During pretraining, the embedding matrix is typically optimized end-to-end to improve generation quality (Li et al., 2022b), as this allows the embeddings to better align with the denoising objective compared to using a frozen embedding matrix from a pretrained language model. However, without additional constraints, the embedding space can exhibit pathological behaviors such as collapse or poor token separation. To address this, recent work has proposed regularization techniques—such as anchor loss and likelihood-aware training—to preserve meaningful structure in the embedding space (Gong et al., 2022; Gao et al., 2024; Gulrajani & Hashimoto, 2024).

### 3.1 EMBEDDING-SPACE SCORE DISTILLATION

Following the practice adopted in latent diffusion (Rombach et al., 2022), we freeze the pretrained embedding matrix during distillation, leaving end-to-end embedding finetuning as a promising direction for future work. While various objectives are possible, we build our method upon Score identity Distillation (SiD; Zhou et al., 2024c) to demonstrate the potential of one-step diffusion models

in the language domain. SiD is a state-of-the-art one-step diffusion distillation method that operates in a fully data-free setting and readily supports two key enhancement techniques—classifier-free guidance (CFG) (Zhou et al., 2024b) and adversarial training (Zhou et al., 2025b)—both of which are found to be important for distillation in the embedding space of DLMs.

Specifically, we denote the pretrained teacher DLM as $\phi$, the student generator as $\theta$, and the score estimator for the student model as $\psi$. Let $E$ denote the token embedding layer and $e \in \mathbb{R}^{d \times L}$ denote the $d$-dimensional continuous embeddings of a sequence of length $L$, which may optionally be mapped back to discrete tokens via a rounding or decoding mechanism during inference. The generation process of the student model is given by

$$e = G_\theta(c, z), \quad z \sim \mathcal{N}(0, \mathbf{I}),$$

where $c$ is an optional condition (*e.g.*, a prompt or label), and $z$ is noise input. We apply the forward diffusion process to obtain noisy embeddings $e_t = \alpha_t e + \sigma_t \epsilon$, $\epsilon \sim \mathcal{N}(0, \mathbf{I})$, where $\alpha_t$ and $\sigma_t$ follow a predefined noise schedule that gradually decreases the signal-to-noise ratio $\alpha_t/\sigma_t$ as $t$ increases.

The pretrained teacher model $\phi$ provides an estimate of the score function at $e_t$ given $t$ and $c$, defined as $s_\phi(e_t, t, c) = \nabla_{e_t} \log p(e_t \mid t, c)$. The distillation objective is to train the student generator such that its score matches that of the teacher in the forward-diffused noisy space. This is achieved by minimizing the model-based explicit score matching (MESM) loss, a form of Fisher divergence:

$$\mathcal{L}_{\text{mesm}}(\theta; \psi^*) = \mathbb{E}_{e=G_\theta(c,z),\, t,\, c,\, z} \left[ \omega_t \left\| s_\phi(e_t, t, c) - s_{\psi^*(\theta)}(e_t, t, c) \right\|^2 \right], \tag{1}$$

where $\psi^*(\theta)$ denotes the true score function induced by the student generator $\theta$, and $\omega_t$ is a time-dependent reweighting coefficient. For unconditional generation, the condition $c$ is set to $\emptyset$.

By Tweedie's formula (Robbins, 1992; Efron, 2011), Equation 1 can be equivalently written as:

$$\mathbb{E}_{e,t,c} \left( \omega_t \frac{\alpha_t^2}{\sigma_t^4} \| \hat{e}_\phi(e_t, t, c) - \hat{e}_{\psi^*(\theta)}(e_t, t, c) \|^2 \right), \tag{2}$$

where $\hat{e}_\phi$ and $\hat{e}_{\psi^*(\theta)}$ denote the expected values of the clean embedding $e$ conditioned on the noisy observation $e_t$, as inferred by the teacher and optimal student score networks, respectively.

While Equation 2 and its gradient are generally intractable to compute, the SiD method (Zhou et al., 2024c) provides an effective optimization procedure that alternates between estimating $\psi^*(\theta)$ and updating $\theta$. Specifically, we optimize $\psi$ given $\theta$ using the denoising score matching (DSM) loss:

$$\mathcal{L}_{\text{dsm}}(\psi) = \mathbb{E}_{e,t,c} \left[ \gamma_t \left\| \hat{e}_\psi(e_t, t, c) - e \right\|^2 \right], \tag{3}$$

and optimize $\theta$ given $\psi$ using the following SiD loss:

$$\mathcal{L}_{\text{sid}}(\theta; \psi^*, \mu) = \mathbb{E}_{e,t,c} \big[ (1-\mu)\, \omega_t \frac{\alpha_t^2}{\sigma_t^4} \left\| \hat{e}_\phi(e_t, t, c) - \hat{e}_\psi(e_t, t, c) \right\|^2$$
$$+ \omega_t \frac{\alpha_t^2}{\sigma_t^4} \left( \hat{e}_\phi(e_t, t, c) - \hat{e}_\psi(e_t, t, c) \right)^\top \left( \hat{e}_\psi(e_t, t, c) - e \right) \big], \tag{4}$$

where $\mu > 0$ is a hyperparameter that is often set as 1 or 1.2.

## 3.2 Adversarial Regularization

While data-free distillation of pretrained diffusion models is appealing—requiring access only to the teacher model rather than real data—and has achieved highly competitive performance in the vision domain (Zhou et al., 2024c;b), its application to DLMs presents a major challenge: degeneration in the student model. In the absence of explicit constraints (*e.g.*, on sentence length) or implicit supervision from real data, distilled models tend to degenerate after a certain number of training iterations, such as (1) generating repetitive tokens, or (2) producing empty sequences filled with `[PAD]` tokens. To mitigate this, we combine standard score distillation with adversarial regularization.

Specifically, when updating the fake score estimator $\psi$, we first sample a condition $c^{\text{fake}}$ and generate an embedding sequence $e_\theta^{\text{fake}}$ using the student generator $\theta$. We then compute the DSM loss of $\psi$ along with part of the adversarial loss—namely, the binary cross-entropy (BCE) loss using pseudo-labels set to all negatives. Additionally, we sample a pair consisting of a real data sequence $x^{\text{real}}$ and its corresponding condition $c^{\text{real}}$, and compute the remaining part of the adversarial loss using pseudo-labels set to all positives. Following Diffusion GAN (Wang et al., 2022) to perform discrimination on noised embeddings, the adversarial loss for $\psi$ is given by:

$$\mathcal{L}_{\text{adv}}^{\text{sg}}(\psi) = \frac{1}{2} \mathbb{E} \left[ \log \sigma(D_\psi(e_t^{\text{real}}, t, c^{\text{real}})) + \log(1 - \sigma(D_\psi(e_{\theta,t}^{\text{fake}}, t, c^{\text{fake}}))) \right], \tag{5}$$

---

**Algorithm 1** DLM-One Adversarial Score Distillation

---

**Input:** Pre-trained teacher DLM $\phi$, student model $\theta$, score estimator $\psi$, embedding layer $E$, score distillation loss coefficient $\mu$, real dataset $\mathcal{D}_{X,C}$, time range $[t_{\min}, t_{\max}]$, diffusion weight function $\lambda(t)$, loss term coefficients $a^{sg}_{dsm}, b^{sg}_{adv}, a^{g}_{sd}, b^{g}_{adv}$.

**Initialization** $\theta \leftarrow \phi, \psi \leftarrow \phi$

**repeat**

    Sample $c^{\text{fake}} \sim \mathcal{D}_{*,Y}$, $(x^{\text{real}}, c^{\text{real}}) \sim \mathcal{D}_{X,C}$, $t \in [t_{\min}, t_{\max}]$

    Sample $z \sim \mathcal{N}(0, \mathbf{I})$, let $e^{\text{fake}} = G_\theta(c^{\text{fake}}, z)$ and $e^{\text{real}} = E(x^{\text{real}})$

    Sample noises $\epsilon^{\text{fake}}, \epsilon^{\text{real}} \sim \mathcal{N}(\mathbf{0}, \mathbf{I})$

    $e^{\text{fake}}_t \leftarrow \alpha_t e^{\text{fake}} + \sigma_t \epsilon^{\text{fake}}, e^{\text{real}}_t \leftarrow \alpha_t e^{\text{real}} + \sigma_t \epsilon^{\text{real}}$

    Compute $\hat{\mathcal{L}}_{\text{dsm}}$ according to Eq. 3 and $\hat{\mathcal{L}}^{sg}_{\text{adv}}$ according to Eq. 5

    Update $\psi$ via SGD on the combined loss $a^{\text{sg}}_{\text{dsm}}\hat{\mathcal{L}}_{\text{dsm}} + b^{\text{sg}}_{\text{adv}}\hat{\mathcal{L}}^{\text{sg}}_{\text{adv}}$

    Sample $c^{\text{fake}} \sim \mathcal{D}_{*,C}$, $t \in [t_{\min}, t_{\max}]$

    Sample $z \sim \mathcal{N}(0, \mathbf{I})$, let $e^{\text{fake}} = G_\theta(c^{\text{fake}}, z)$

    Sample noises $\epsilon^{\text{fake}} \sim \mathcal{N}(\mathbf{0}, \mathbf{I})$

    $e^{\text{fake}}_t \leftarrow \alpha_t e^{\text{fake}} + \sigma_t \epsilon^{\text{fake}}$

    Compute $\hat{\mathcal{L}}_{\text{sd}}$ according to Eq. 4 and $\hat{\mathcal{L}}^{g}_{\text{adv}}$ according to Eq. 6

    Update $\theta$ via SGD on the combined loss $a^{\text{g}}_{\text{sd}}\hat{\mathcal{L}}_{\text{sd}} + b^{\text{g}}_{\text{adv}}\hat{\mathcal{L}}^{\text{g}}_{\text{adv}}$

**until** the maximum number training steps is reached

**Output:** $\theta$

---

where $e^{\text{real}}_t$ is the noisy embedding obtained by forward diffusing the embedding of $x^{\text{real}}$. For the update steps of the student model $\theta$, we compute both the SiD loss and the all-positive BCE loss on generated sequences conditioned on $c$. We denote each generated $\langle \text{data}, \text{condition} \rangle$ pair as $(x_\theta, c)$ and $e_{\theta,t}$ as the noised version of of $e_\theta$. The corresponding adversarial loss is:

$$\mathcal{L}^{\text{g}}_{\text{adv}}(\theta) = \mathbb{E}\left[\log \sigma(D_\psi(e_{\theta,t}, t, c))\right]. \tag{6}$$

We provide an overview and pseudo-code of our adversarial score distillation training process in Figure 1 and Algorithm 1, respectively. For efficiency, we utilize the same model (*i.e.,* the score estimator $\psi$) for both score prediction and GAN discrimination. At a high level, the additional adversarial losses provide implicit supervision and help stabilize training, preventing mode collapse and encouraging more realistic sequence generation.

### 3.3 TWO-STAGE TRAINING

Due to the alternating update scheme, the score estimator $\psi$ may fail to provide an accurate approximation of the true score corresponding to the student model $\theta$. To address this issue, we propose a two-stage training procedure. In the first stage (Stage 1), our primary goal is to obtain a "good enough" student model whose generative distribution is reasonably close to that of the teacher. This can be assessed using standard performance metrics such as BLEU. In practice, we train the student model for a fixed number of steps and select the best checkpoint based on BLEU score evaluated on the validation set.

In the second stage (Stage 2), we resume training the student model $\theta$ from the selected checkpoint but reinitialize the score estimator $\psi$ with the parameters of the teacher model $\phi$. The intuition behind this is to mitigate the potential lag of $\psi$, which arises because it is updated alternately with the student and may fall behind the true score of the evolving student model. This issue becomes more pronounced as the student's generative distribution grows increasingly close to the teacher's, diverging significantly from its earlier state. In such cases, the feedback provided by $\psi$ may become insufficient to guide further improvement. Reinitializing $\psi$ with the teacher model helps realign it with the updated student and provides more meaningful learning signals for continued distillation. The Stage 2 training procedure largely mirrors that of Algorithm 1, with the key distinction that it requires a student model checkpoint from the end of Stage 1 for initialization.

## 4 EXPERIMENTS

In our experiments, we conduct a comprehensive evaluation on the benchmark tasks originally used to assess the performance of the teacher DLMs pretrained with DiffuSeq and Plaid. The results convincingly demonstrate the potential of significantly accelerating the sampling efficiency of

Table 1: **Performance comparison between teacher and student models across Seq2Seq tasks.** ↑ indicates higher is better, ↓ indicates lower is better. * denotes that the student's performance is within 5% of the teacher's, and ** indicates that it is within 1%.

| Task | Model | BLEU(↑) | ROUGE-L(↑) | BERT(↑) | Dist-1(↑) | SelfBLEU(↓) / Div-4(↑) | NFEs(↓) |
|------|-------|---------|-----------|---------|-----------|------------------------|---------|
| PP | DiffuSeq | 0.1829 | 0.5299 | 0.7932 | 0.9747 | 0.2732 / 0.8641 | 2000 |
|    | DLM-One | 0.1788* | 0.5265** | 0.7851* | 0.9671** | 0.3418 / 0.6256 | **1** |
| QG | DiffuSeq | 0.1512 | 0.3468 | 0.5871 | 0.9141 | 0.2789 / 0.8103 | 2000 |
|    | DLM-One | 0.1512** | 0.3257 | 0.5683* | 0.9053** | 0.6166 / 0.3798 | **1** |
| TS | DiffuSeq | 0.2929 | 0.5313 | 0.7781 | 0.9272 | 0.4642 / 0.6604 | 2000 |
|    | DLM-One | 0.2927** | 0.5299** | 0.7565* | 0.8924* | 0.5456 / 0.4098 | **1** |

continuous DLMs via score distillation, enabling one-step token sequence generation that rivals the performance of teacher models requiring hundreds of times more computation. This redefines the Pareto frontier between computational efficiency and generation quality in continuous diffusion-based language modeling, and has profound implications for the future development of LLMs.

## 4.1 TASKS AND DATASETS

We consider three sequence-to-sequence (Seq2Seq) tasks, including: question generation (QG), text simplification (TS), and paraphrase (PP). Specifically, we used preprocessed data from Quasar-T (Dhingra et al., 2017) for QG, Wiki-Auto (Jiang et al., 2020) for TS, and Quora question pairs (QQP) for PP. For each dataset, we use the standard splits of training, validation, and test sets. The data derived from Quasar-T contain approximately 129k ⟨document, question⟩ pairs, including 117k training pairs, 2k validation pairs, and 10k test pairs. The Wiki-Auto preprocessed dataset consists of a total of ∼685k ⟨complex, simple⟩ sentence pairs, with approximately 678k training pairs, 2k validation pairs, and 5k test pairs. QQP dataset contains about 150k paraphrase sentence pairs, including 145k training, 2k validation, and 2k test. In addition to Seq2Seq tasks, we also conduct experiments on unconditional text generation. For this setting, we follow the setup introduced by Gulrajani & Hashimoto (2023) and use the OpenWebText2 (Gao et al., 2020) dataset, which consists of high-quality English web content filtered to resemble the pretraining corpus of GPT-2. The dataset contains approximately 10 million documents, and we use a 1B-token subset for training, consistent with prior work.

## 4.2 EVALUATION

For evaluation of the Seq2Seq tasks, we mainly consider five factors: BLEU (Papineni et al., 2002), ROUGE-L (Lin, 2004), BERT Score (Zhang et al., 2020), Dist-1, and sequence diversity. BLEU, ROUGE-L, and BERTScore are standard metrics for evaluating sequence-to-sequence tasks, as they capture sentence-level similarity between the generated sequences and the references. BLEU emphasizes n-gram precision, ROUGE-L focuses on recall based on the longest common subsequence, and BERTScore leverages contextual embeddings to assess semantic similarity. Dist-1 measures lexical diversity by computing the average ratio of distinct unigrams in a single sentence over all generated samples. Sequence-level diversity is further assessed using two metrics: self-BLEU (Zhu et al., 2018) and Div-4. Following the implementation of DiffuSeq (Gong et al., 2022), we compute self-BLEU by averaging inter-sentence BLEU scores across generated samples, while Div-4 quantifies the proportion of distinct 4-grams among them. For the unconditional generation task, we assess the model performance mainly through the generative perplexity. Specifically, we generate 250 samples and calculate the average perplexity evaluated under GPT-2 (Radford et al., 2019).

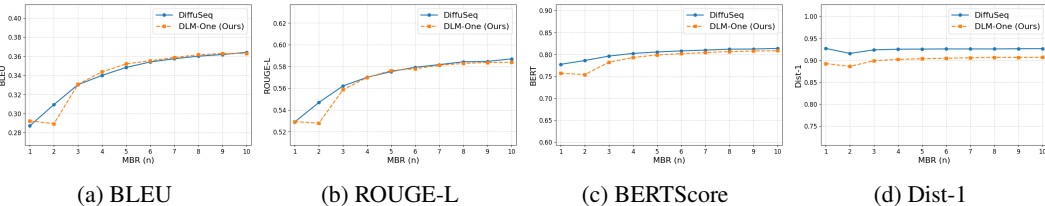

(a) BLEU     (b) ROUGE-L     (c) BERTScore     (d) Dist-1

Figure 2: Evaluation metrics using MBR decoding across 1 to 10 candidate(s) on the Wiki dataset.

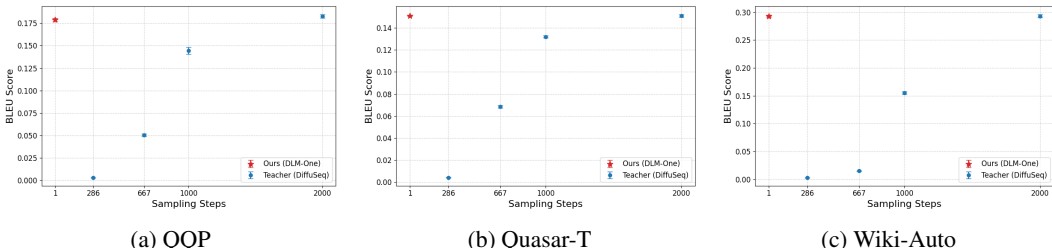

(a) QQP          (b) Quasar-T          (c) Wiki-Auto

Figure 3: **BLEU score vs. sampling steps on different datasets.** The teacher model (DiffuSeq) requires hundreds to thousands of denoising steps to reach optimal performance, while our DLM-One achieves competitive BLEU in a single step—offering over **100× faster** generation without significant quality degradation.

Table 2a: **Average generation time per sample across different sampling steps of DiffuSeq.** Each entry reflects the mean time (in seconds), averaged over 100 runs. Time does not scale strictly linearly with NFEs, due to fixed overhead such as embedding rounding and tokenizer-based decoding.

Table 2b: **Mean perplexity of the generated samples output by DLM-One (Student) and Plaid (Teacher) using different inference steps.** The results correspond to the unconditional text generation task evaluated under GPT-2.

| Steps | 1 | 65 | 286 | 667 | 1000 | 2000 |
|---|---|---|---|---|---|---|
| **Time (s)** | 0.03 | 0.51 | 2.25 | 5.20 | 7.70 | 14.94 |

| Model | Student | Teacher | | | |
|---|---|---|---|---|---|
| **# Inf. Steps** | 1 | 16 | 64 | 256 | 4096 |
| **Perplexity** | 93.99 | 298.92 | 122.41 | 94.28 | **83.37** |

### 4.3 Sequence-to-Sequence (Seq2Seq) Tasks

For sequence-to-sequence tasks, we mainly consider DiffuSeq (Gong et al., 2022) as our major baseline to showcase the effectiveness of the proposed score distillation framework for LMs. We list results of all five performance metrics in Table 1, which shows that our distilled models can achieve close-to-teacher performance consistently across all three tasks while taking far less number of functional evaluations (NFEs). The actual acceleration is further demonstrated in Figure 3, where we consider the BLEU score against number of sampling steps on QQP, Quasar-T, and Wiki-Auto datasets. In Table 2a, we provide the conversion from the sampling steps to the inference time, which is measured on an NVIDIA RTX A5000 GPU. Our one-step model achieves up to an approximately **500×** speedup compared to the 2000-step baseline with no notable performance degradation.

The results of our approach on PP and QG are obtained from the final-stage (i.e., Stage 2) DLM-One models, while those on TS are reported from Stage 1, as the student model already closely matches the teacher's performance. As shown in Figure 2, minimum Bayes risk (MBR) decoding offers a more comprehensive evaluation of generation quality and diversity by leveraging multiple candidate samples. As the number of candidates increases, MBR decoding typically leads to improved performance. The observation that our student model consistently matches the teacher across 1 to 10 candidates under MBR decoding further suggests that a single-stage distillation is sufficient for the TS task on the Wiki dataset.

### 4.4 Unconditional Text Generation

To assess the generality and scalability of DLM-One, we further conduct experiments on a more complex text generation task and a larger scale dataset. Specifically, we select Plaid (Gulrajani & Hashimoto, 2024)as our baseline for unconditional text generation on the OpenWebText2 dataset. We configure the teacher Plaid model to use a 16-block transformer backbone with 384 hidden dimensions and 6 attention heads. As shown in Table 2b, our DLM-One model achieves competitive performance compared to multistep Plaid teacher model up to 256 inference steps, which is effectively a 256× speedup.

## 5 Discussion

**Analysis of Model Degeneration.** When directly applying data-free diffusion distillation methods for vision models to DLMs, we noticed that the student model will inevitably suffer from inferior

Table 3: **Effect of two-stage training on the QQP dataset.** The second row shows raw scores; the third row shows relative changes from Stage 1. Percentages in green and red indicate improvements and degradations, respectively. Arrows ↑/↓ denote preferred directions.

| Stage | BLEU(↑) | ROUGE-L(↑) | BERT(↑) | Dist-1(↑) | SelfBLEU(↓) | Div-4(↑) |
|---|---|---|---|---|---|---|
| Stage 1 | 0.1468 | 0.4829 | 0.7402 | 0.9370 | 0.2195 | 0.7764 |
| Stage 2 | 0.1788 | 0.5265 | 0.7851 | 0.9671 | 0.3418 | 0.6256 |
| ΔStage | +21.8% | +9.0% | +6.1% | +3.2% | +55.7% | −19.4% |

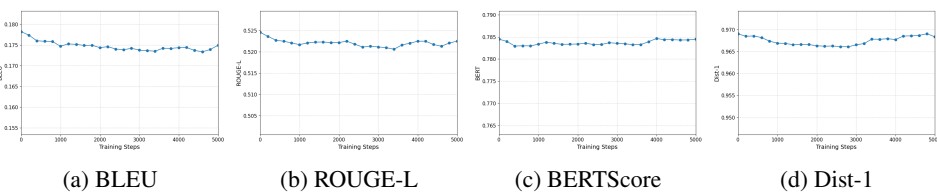

| (a) BLEU | (b) ROUGE-L | (c) BERTScore | (d) Dist-1 |

Figure 4: Evolution of evaluation metrics during Stage 3 training on the QQP dataset.

generation quality and even degeneration. After thoroughly reviewing the failure patterns and inspecting the deeper causes, we identify two major issues specific to DLM distillation:

1. **Poor Initial Predictions.** Unlike in the vision domain, where diffusion model's predictions at a large timestep can be blurry but still recognizable, a DLM's initial predictions are often incoherent (*e.g.,* gibberish and making little sense), exacerbating the inaccurate score estimation problem of the fake score network and restricting the student's final performance (see also Section 3.3). Therefore, in DLM-One, we propose a two-stage training approach to mitigate this initial score mismatch issue in the second stage.

2. **Variable Output Length.** Unlike multistep models, one-step DLMs must determine the final output length upfront, which makes training unstable and prone to degeneration. One specific issue is related to the use of trailing `[PAD]` tokens. Without supervision from data, the student generator can easily fail by simply learning to output a target sequence full of `[PAD]` tokens, because it can trick the teacher model as a "valid" data pattern and lead to very small discrepancy between the teacher model and the fake score model. This issue is also noted in Section 3.2, where we introduce an adversarial loss term to provide regularization on the generation sequence length.

**Effect of Two-stage Training.** We find the two-stage training strategy is crucial for improving the model's overall fidelity across key metrics such as BLEU, ROUGE-L, and BERTScore, at the cost of reduced diversity. For practical applications of DLM-One, we argue this is a favorable trade-off, as higher fidelity often corresponds to greater model utility for end-users. The trade-off is further quantified in Table 3, which compares the performance of the QQP checkpoints from the two stages. Furthermore, we demonstrate that this loss in diversity can be mitigated with inference-time augmentation, and we also discuss the possibility of generalizing DLM-One to a few-step model, as detailed in Appendix C.

**Limited Gain from Additional Stages.** A natural question arises: *Will more stages continue to improve performance?* Based on our experiments, the answer appears to be no. As illustrated in Figure 4, model performance essentially plateaus at the beginning of a third stage, and while minor fluctuations are observed thereafter, the metrics do not exhibit new upward trends. Further training does not yield additional gains, likely due to diminishing learning signals.

**Effect of Additional Steps at Inference Time.** Although DLM-One is optimized for single-step generation, we explore whether introducing additional steps at inference time can further enhance generation quality. Specifically, we implement a simple iterative scheme in which the model alternates between re-noising and denoising its own output multiple times. As shown in Table 4, increasing the number of steps from 1 to 4 consistently improves fidelity metrics (*e.g.,* BLEU, ROUGE-L) at a modest cost to diversity (*e.g.,* Div-4, self-BLEU). This demonstrates that even without explicit multi-step training, the number of sampling steps can serve as a practical lever to navigate the quality-diversity trade-off. However, since the model was not optimized for this regime, these results should not be interpreted as an upper bound. Training distilled generators specifically for few-step

Table 4: Performance of DLM-One under increased inference steps on the QQP dataset.

| Steps | BLEU(↑) | ROUGE-L(↑) | BERT(↑) | Dist-1(↑) | SelfBLEU(↓) | Div-4(↑) |
|---|---|---|---|---|---|---|
| 1 | 0.1788 | 0.5265 | 0.7851 | 0.9671 | **0.3418** | **0.6256** |
| 2 | 0.1800 | 0.5287 | 0.7895 | 0.9676 | 0.3455 | 0.6228 |
| 4 | **0.1829** | **0.5329** | **0.7959** | **0.9693** | 0.3549 | 0.6095 |

inference, a promising direction inspired by recent vision models (Yin et al., 2024; Zhou et al., 2025a), remains a key avenue for future work.

**Comparison with AR Models.** A key advantage of our approach is that DLM-One maintains competitive generation quality compared to similar-sized AR LMs while offering a substantial improvement in inference speed. Our results align with prior work, which shows that teacher DLMs (*e.g.,* DiffuSeq, 91M parameters) can already achieve on-par or superior performance to much larger, fine-tuned AR models like GPT-2 Large (774M parameters) across tasks such as paraphrase, question generation, and text simplification. By distilling the teacher into a one-step generator, DLM-One preserves this high fidelity while being orders of magnitude faster than both its multi-step teacher and the token-by-token sampling of AR models. We provide a detailed comparison of performance metrics, model sizes, and inference speeds in Appendix D.

## 6 CONCLUSION

In this work, we propose a practical distillation framework for training continuous diffusion language models for one-step sequence generation (DLM-One), eliminating the need for iterative refinement during generation. Our method is broadly applicable to continuous diffusion-based language models and enables fast, one-step generation via score distillation from pretrained teacher models. To further stabilize training and improve student quality, we introduce a two-stage training scheme with adversarial regularization. Through detailed experiments on conditional text generation tasks, we demonstrate that DLM-One achieves competitive performance against the teacher DLMs while reducing sampling cost by up to $\sim500\times$. This redefines the Pareto frontier between computational efficiency and generation quality in continuous diffusion-based language modeling, and has profound implications for the future development of LLMs.

Nevertheless, our work opens up several promising directions for future investigation. First, while hyperparameters like the score distillation loss coefficient $\mu$ currently require per-task tuning, future work could explore more principled and adaptive training schemes. Second, we find that the trade-off between fidelity and diversity in DLM-One is a controllable aspect that can be adjusted based on the needs of downstream applications. We identify the extension of DLM-One to a few-step generator could be a key avenue for future research, with the potential to improve the overall model performance in both fidelity and diversity.

REPRODUCIBILITY STATEMENT

To ensure the reproducibility of our work, we provide detailed implementation and training protocols in Appendix B. This includes all distillation-related hyperparameters for each dataset (Table 5) and a description of the conditioning mechanism. Our work is based on the publicly available DiffuSeq codebase, which we have linked in the appendix. We will release our full source code, including scripts to reproduce all experiments and a link to our final model checkpoints, upon publication.

ETHICS STATEMENT

This work introduces one-step sequence generation framework, DLM-One, to significantly improve the inference speed of DLMs. The primary goal is to reduce the high computational cost and energy consumption associated with large generative models, thereby making this technology more accessible and sustainable, as discussed in our broader impacts statement in Appendix A. We used publicly available, standard benchmark datasets (QQP, Quasar-T, Wiki-Auto) for our experiments. While our work makes generation more efficient, it does not introduce new risks beyond those inherent in existing language models, such as the potential for generating biased or harmful content. We believe the net impact of this research is positive, as it contributes to more computationally efficient and environmentally friendly AI.

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

# Appendix for DLM-One

## A  BROADER IMPACTS

The high computational cost of large-scale language models poses challenges for accessibility, especially for users with limited resources. DLM-One addresses this by enabling one-step diffusion-based language generation, offering a significantly more efficient alternative to traditional iterative methods. By reducing the number of function evaluations required at inference time, DLM-One lowers energy consumption and makes diffusion language models more practical and sustainable for real-world deployment.

## B  IMPLEMENTATION DETAILS

In this section, we provide detailed documentation of the implementation, including aspects not fully covered in the main text, for experiments on DiffuSeq. We outline the specific adaptations required for distilling these baselines into one-step sequence generators.

### B.1  DIFFUSEQ

We adopt the official codebase of DiffuSeq[1] and all three released checkpoints to conduct our Seq2Seq experiments in Section 4.

#### B.1.1  TRAINING PROTOCOL

For the training of our DLM-One student models, we set a fixed training budgets of 50,000 steps for all datasets. We use AdamW (Loshchilov & Hutter, 2019) optimizer with $\beta_1 = 0.0$, $\beta_2 = 0.999$, and zero weight decay for both the student and the score estimator. The learning rate is fixed across tasks at $10^{-5}$. During Stage 1, we monitor the performance metrics on the validation set, such as BLEU, every 200 steps. Once the training is completed, we select the best-performing student checkpoint on the validation set as our new starting point for Stage 2. We provide a detailed table of distillation-related hyperparameter for both stages of each dataset in Table 5.

Table 5: Distillation-related hyperparameters used in Stage 1 and Stage 2 across different datasets.

| Dataset | Stage | $\mu$ | $[t_{\min}, t_{\max}]$ | $t_{\text{init}}$ | $a^{sg}_{dsm}, b^{sg}_{adv}$ | $a^{g}_{sd}, b^{g}_{adv}$ | $\text{lr}_\psi$ | $\text{lr}_\theta$ |
|---|---|---|---|---|---|---|---|---|
| QQP | Stage 1 | 1.2 | $[0, 1976]$ | 1490 | 0.5, 0.5 | 0.5, 0.5 | 3e-5 | 1e-5 |
|     | Stage 2 | 0.5 | $[0, 1976]$ | 1490 | 0.5, 0.5 | 0.9, 0.1 | 1e-5 | 1e-5 |
| Q-T | Stage 1 | 1.2 | $[0, 1976]$ | 1490 | 0.5, 0.5 | 0.5, 0.5 | 1e-5 | 1e-5 |
|     | Stage 2 | 1.2 | $[0, 1976]$ | 1490 | 0.5, 0.5 | 0.5, 0.5 | 1e-5 | 1e-5 |
| Wiki | Stage 1 | 1.0 | $[0, 1976]$ | 1490 | 0.5, 0.5 | 0.5, 0.5 | 1e-5 | 1e-5 |

#### B.1.2  CONDITIONING

During the pretraining of DiffuSeq models, the injection of conditions is achieved via concatenation, *i.e.,* the condition sequence is directly concatenated with the data sequence as a whole before entering the network. However, the positions corresponding to the condition sequence do not participate in the diffusion forward process and are output as-is by the models. To align with the teacher pretraining process, we adjust the output by the student model accordingly. Denote the condition embedding sequence as $e^{\text{cond}}$ and the initial noise for the student model $\theta$ as $z$. Let $\tilde{e}_{\theta,t} = G_\theta(e^{\text{cond}}, z) = e^{\text{cond}}_\theta \oplus e^{\text{data}}_\theta$. To inject the true condition, we modify the direct output by the student model (*i.e.,* $\tilde{e}_{\theta,t}$) as $e_{\theta,t} = e^{\text{cond}} \oplus e^{\text{data}}_\theta$. The rationale behind this operation is that the teacher model has been trained on the true conditions from the real dataset only, using part of the generated sequence would introduce a discrepancy between teacher pretraining and distillation. Therefore, we replace the generated condition part, *i.e.,* $e^{\text{cond}}_\theta$ with the true condition sequence $e^{\text{cond}}$. In our early experiments, we found that this adjustment helps stabilize training and preventing degeneration when used together with adversarial training.

---

[1] https://github.com/Shark-NLP/DiffuSeq

## C   ON THE FIDELITY-DIVERSITY TRADE-OFF

The empirical results of DLM-One model reflect a trade-off between generation fidelity and diversity. The proposed two-stage training process tends to prefer high-fidelity outputs over high-diversity outputs, which we argue is a not necessarily a limitation for practical applications. In this section, we provide a detailed analysis of this trade-off and suggest how one can manage the trade-off in practice: 1) inference-time text augmentation to boost diversity, and 2) few-step generalization to increase DLM-One's overall performance.

### C.1   MITIGATING DIVERSITY LOSS WITH TEXT AUGMENTATION

High fidelity is often preferable in practice, as many users want to call a model once and receive a high-quality, relevant answer. Our model is optimized for this single-call scenario, where higher fidelity metrics (*e.g.,* BLEU, BERTScore) indicate stronger utility. While the resulting decrease in diversity might seem like a limitation, we demonstrate that it can be compensated for at inference time with a simple rule-based text augmentation.

Specifically, by randomly inserting a pad token (`[PAD]`) into the condition text with a given probability, we can directly boost the diversity of the generated sentences. Table 6 presents the results of this experiment on PP, QG, and TS tasks. As the insertion probability increases, diversity metrics like Div-4 consistently rise across all tasks, accompanied by a predictable, modest decrease in fidelity scores. This shows that diversity in DLM-One is not a fixed limitation but rather a controllable parameter that can be tuned according to the needs of a specific application. We anticipate that with more advanced techniques, such as model-based augmentation, generation diversity could be further enhanced with even less impact on fidelity.

Table 6: The effect of random `[PAD]` token insertion on the fidelity-diversity trade-off across three tasks. As insertion probability increases, diversity (SelfBLEU and Div-4) consistently improves at the cost of fidelity.

| Task | Dataset | Ins. Prob. | BLEU($\uparrow$) | R-L($\uparrow$) | BERT($\uparrow$) | Dist-1($\uparrow$) | SelfB($\downarrow$) / Div-4($\uparrow$) |
|------|---------|-----------|---------|--------|---------|----------|------------------|
| PP | QQP | 0.0 | 0.1788 | 0.5265 | 0.7851 | 0.9671 | 0.3418 / 0.6256 |
|    |     | 0.5 | 0.1746 | 0.5204 | 0.7798 | 0.9663 | 0.3224 / 0.6507 |
|    |     | 0.7 | 0.1712 | 0.5177 | 0.7771 | 0.9654 | 0.3134 / 0.6608 |
| QG | Q-T | 0.0 | 0.1512 | 0.3257 | 0.5683 | 0.9053 | 0.6166 / 0.3798 |
|    |     | 0.5 | 0.1485 | 0.3175 | 0.5632 | 0.9065 | 0.5820 / 0.4167 |
|    |     | 0.7 | 0.1473 | 0.3144 | 0.5624 | 0.9064 | 0.5692 / 0.4294 |
| TS | Wiki | 0.0 | 0.2927 | 0.5299 | 0.7565 | 0.8924 | 0.5456 / 0.4098 |
|    |      | 0.5 | 0.2769 | 0.5196 | 0.7486 | 0.8897 | 0.5015 / 0.4532 |
|    |      | 0.7 | 0.2715 | 0.5166 | 0.7464 | 0.8890 | 0.4866 / 0.4665 |

### C.2   IMPROVING DIVERSITY WITH MULTI-STEP TRAINING

A more fundamental approach to improving both fidelity and diversity is to train the model for few-step generation. This would involve training the generator to perform denoising at different noise levels (*i.e.,* step-aware training), significantly improving the model's ability to produce diverse results. Such strategies have proven highly effective for enhancing distilled student models in the vision domain (Salimans et al., 2024; Zhou et al., 2025a) and represent a promising direction for future work on continuous DLMs.

## D   COMPARISON WITH AUTOREGRESSIVE (AR) MODELS

To evaluate the performance and efficiency of DLM-One against standard baselines, we compare it to its teacher model (DiffuSeq) and two fine-tuned autoregressive models (GPT-2 Base and GPT-2 Large). For inference speed, we report the average time in seconds over 100 runs on the text simplification task, with a maximum output length of 128 tokens for a fair comparison.

The results, shown in Table 7, demonstrate a clear trade-off between model type, performance, and speed. Both DiffuSeq and our distilled DLM-One are competitive with or outperform the GPT-2 models on key fidelity metrics (BLEU, ROUGE-L, BERT), despite having significantly fewer

parameters than GPT-2 Large. Most notably, DLM-One's single-step generation makes it by far the fastest model, achieving a speedup of approximately **27**× over GPT-2 Base and **500**× over its teacher, DiffuSeq.

Table 7: **Comparison of DLMs and AR models on the text simplification task.** DLM-One maintains competitive performance while being orders of magnitude faster. All results are reported using MBR-10 decoding for a fair comparison.

| Model | BLEU(↑) | R-L(↑) | BERT(↑) | Dist-1(↑) | SelfB(↓) | Div-4(↑) | # Params | Avg. Inf. Time (s) |
|---|---|---|---|---|---|---|---|---|
| GPT-2 Base FT | 0.3083 | 0.5461 | 0.8021 | 0.9439 | 0.5444 | 0.6047 | 117M | 0.82 |
| GPT-2 Large FT | 0.2693 | 0.5111 | 0.7882 | **0.9464** | 0.6042 | 0.5876 | 774M | 2.34 |
| DiffuSeq (Teacher) | 0.3622 | **0.5849** | **0.8126** | 0.9264 | **0.4642** | **0.6604** | **91M** | 14.94 |
| DLM-One (Student) | **0.3630** | 0.5839 | 0.8084 | 0.9068 | 0.5456 | 0.4098 | **91M** | **0.03** |

## E  ADDITIONAL RESULTS

Due to the page limit of the main text, we defer supplementary experimental results to this section.

### E.1  GENERATED SAMPLES FOR SEQ2SEQ TASKS

We present generation results on 5 random examples each from the PP, QG, and TS tasks in Tables 9 to 11.

### E.2  DLM-ONE WITH MBR DECODING

To directly compare with the results reported in Gong et al. (2022), we evaluate our student models using the MBR decoding strategy with a total of 10 generated candidates (denoted as MBR-10). As shown in Table 8, our distilled models demonstrate comparable performance to their respective teachers across all three datasets (QQP, QG, Wiki). In particular, the student model on the Wiki dataset nearly matches the teacher in all quality metrics (BLEU, ROUGE-L, BERTScore), suggesting that the DLM-One model can retain strong performance even when evaluated using multiple samples. However, we also observe a decrease in diversity metrics, especially on QG, which indicates that MBR may favor models with higher inter-sentence diversity.

## F  LLM USAGE STATEMENT

We utilized large language models (*e.g.,* ChatGPT) to assist with proofreading, grammatical corrections, and polishing the text of this manuscript. No new scientific results or text contents were generated by LLMs.

Table 8: **MBR-10 evaluation results across Seq2Seq tasks.** Arrows indicate preferred directions: ↑ higher is better, ↓ lower is better.

| Task | Dataset | Model | BLEU(↑) | R-L(↑) | BERT(↑) | Dist-1(↑) | SelfB(↓) / Div-4(↑) |
|------|---------|-------|---------|--------|---------|-----------|---------------------|
| PP | QQP | DiffSeq | 0.2413 | 0.5880 | 0.8365 | 0.9807 | 0.2732 / 0.8641 |
|    |     | DLM-One | 0.2213 | 0.5741 | 0.8297 | 0.9773 | 0.3418 / 0.6256 |
| QG | Q-T | DiffSeq | 0.1731 | 0.3665 | 0.6123 | 0.9056 | 0.2789 / 0.8103 |
|    |     | DLM-One | 0.1522 | 0.3280 | 0.5708 | 0.9026 | 0.6167 / 0.3798 |
| TS | Wiki | DiffSeq | 0.3622 | 0.5849 | 0.8126 | 0.9264 | 0.4642 / 0.6604 |
|    |      | DLM-One | 0.3630 | 0.5839 | 0.8084 | 0.9068 | 0.5456 / 0.4098 |

Table 9: **Examples from the Paraphrase (PP) task.** Each example consists of a source sentence, a reference sentence, and outputs generated by DiffuSeq (Teacher) and DLM-One (Student).

| Source | Reference | Recover | |
|--------|-----------|---------|--|
|        |           | **DiffuSeq** | **DLM-One** |
| how can i be a good geologist? | what should i do to be a great geologist? | how do i really be a good geologist? | how can i become a good geologist? |
| which are the best engineering fields? | what is the best field of engineering? | which are the best engineering field? | what are the best engineering fields? |
| how do i become an attractive girl? | how do you become pretty / attractive? | how can one become a girl? | how can i become an attractive girl quickly? |
| how does a long distance relationship work? | do long distance relationships work? | does long distance relationship work? | how do i have a long distance relationship? |
| what are some interesting things to do when bored? | what should i do if i'm badly bored? | what should you do when you bored? | what are the best thing to do when bored? |

Table 10: **Examples from the Question Generation (QG) task.** Each example consists of a source sentence, a reference sentence, and outputs generated by DiffuSeq (Teacher) and DLM-One (Student).

| Source | Reference | Recover | |
|--------|-----------|---------|--|
| | | **DiffuSeq** | **DLM-One** |
| a gaggle is a group of geese. | what is a group of geese called | what kind of birds would you a group geese geese | what is a group of geese called? |
| the ten - mineral mohs scale of relative hardness, based on what scratches what. | what is measured by moh's scale? | in minerology what does the mohs scale measure | in minerology what does the mohs scale measure |
| if you mix red and green lights they do not magically change into yellow light. | what colour do you get when you mix blue and yellow together? | when you mix equal amounts of blue and yellow color do what color? | when you mix equal amounts of blue and yellow yellow, what color do you get? |
| capable of sustained hovering, the hummingbird has the ability to fly deliberately backwards | which is the only musical bird that can fly backwards | what is the only bird that can can fly backwards | what is the only bird that can fly backwards |
| alexander graham bell in 1876, at the age of 29, alexander graham bell invented his telephone. | what did alexander graham bell invent | the telephone was invented in which year | the telephone was invented in which year |

Table 11: **Examples from the Text Simplification (TS) task.** Each example consists of a source sentence, a reference sentence, and outputs generated by DiffuSeq (Teacher) and DLM-One (Student).

| Source | Reference | Recover | |
|---|---|---|---|
| | | **DiffuSeq** | **DLM-One** |
| she was also the leader of the party between 1993 and 1995. | she was also the leader of the party between 1993 and 1995. | she was the leader of the party from 1995 to 1993. | she was the leader between 1993 and 1995. |
| thiel - sur - acolin is a commune in the allier department in auvergne - rhone - alpes in central france. | thiel - sur - acolin is a commune. | thiel - sur - acolin is a commune. | thiel - sur - acolin is a commune. |
| vetlanda municipality ( " vetlanda kommun " ) is a municipality in jonkoping county, in southern sweden where the town of vetlanda is the seat. | vetlanda municipality is a municipality in jonkoping county in southern sweden. | vetlanda municipality is a municipality in jonkoping county in southern sweden. | vetlanda municipality is a municipality in jonkoping county. |
| beaufort is located in north carolina's " inner banks " region. | beaufort is in north carolina's inner banks region. | beaufort is in north carolina's " inner banks " region. | beaufort is located in " inner banks " region. |
| weaver was born in pittsburgh, pennsylvania, on january 19, 1926, the son of elsa w. ( nee stringaro ) weaver and john carson weaver. | weaver was born on january 19, 1926 in pittsburgh, pennsylvania. | weaver was born in pittsburgh, pennsylvania, on january 19, 1926. | weaver was born in pittsburgh, pennsylvania. |