# OpenReview forum: "DLM-One: Diffusion Language Models for One-Step Sequence Generation"
_ICLR.cc/2026/Conference — Submitted to ICLR 2026_

### Official Review · Reviewer_gjET · 2025-10-31

**Soundness:** 2
**Presentation:** 3
**Contribution:** 2
**Rating:** 6
**Confidence:** 2

**Summary:**

This paper introduces DLM-one, a practical score distillation framework for continuous DLMs that enables one-step sequence generation without iterative denoising. This framework eliminates the need for iterative refinement by aligning the outpt scores of the student model in the continuous token embedding space with the score function of the pre-trained teacher DLM, significantly improving the sampling efficiency.

**Strengths:**

As mentioned in this paper, DLM-One achieves up to 500\times speedup in inference time, which means on the premise of ensuring quality, the computing cost and time consumption can be significantly reduces.

**Weaknesses:**

Although the results are impressive, the paper may not have elaborated in sufficient detail on the contributions of different components to the final performance. For example, the impact of different choices of teacher models.

**Questions:**

1.Since a comparison with the AR model is listed in the Appendix D, how does the DLM-one perform in handling open-domain creative text generation?
2.Is the one-step sequence generation model first proposed in this paper?

---

> ### Author Response · Authors · 2025-11-22
>
> Thank you for your positive feedback. We are encouraged by your comments on the significant inference speedup achieved by DLM-One and its competitive quality. Regarding your concerns and questions, we would like to clarify in the following:
>
> 1. **Insufficient Details on Components' Contributions.**
>     In the paper, we indeed explore the impact of different choices of DLM-One model itself in detail, including the **"Effect of Two-stage training," "Gain from Additional Stages," "Effect of Additional Inference Steps,"** (in **Section 5 "Discussion"**) and **"Necessity of Adversarial Regularization"** (in **Section 3.2 "Adversarial Regularization"**).
>
> 2. **Different Choices of Teacher Models.** We further conducted new experiments on other teacher DLMs such as **Plaid** [1] and **TESS** [2]. Together with the results of DiffuSeq, they demonstrate the effectiveness of DLM-One regardless of the choice of teacher DLM:
>
>     i. **Plaid** as teacher: We pretrained a custom Plaid teacher model on OpenWebText2 and evaluated the GPT-2-based generative perplexity of unconditional generation results using different numbers of inference steps. While the teacher Plaid model can reach a perplexity score of **83.37 using 4096 steps** and **94.28 using 256 steps**, our student DLM-One can achieve a perplexity score of **93.99 using just one step**, which is effectively a **256**$\times$ speedup.
>
>     |# Inference Steps|1 Step|16 Steps|64 Steps|256 Steps|4096 Steps|
>     |:--:|:--:|:--:|:--:|:--:|:--:|
>     |Plaid (Teacher)|-|298.92|122.41|$\underline{94.28}$|**83.37**|
>     |DLM-One-Plaid (Student)|$\underline{93.99}$|-|-|-|-|
>
>     ii. **TESS** as teacher: We followed the official configurations to pretrain a TESS model on Paraphrase (PP) task and QQP dataset, and then distilled the pretrained model by adopting the same principle of DLM-One with TESS-specific adaptation. We would like to share some of our preliminary results here:
>
>     |Model|BLEU|Rouge-L|Bert|Dist-1|# Inference Steps|
>     |:--:|:--:|:--:|:--:|:--:|:--:|
>     |DiffuSeq|0.1829|0.5299|0.7932|0.9747|2000|
>     |DLM-One-DiffuSeq|0.1788|0.5265|0.7851|0.9671|1|
>     |**TESS**|**0.3037**|**0.6182**|**0.9496**|**0.9860**|**1000**|
>     |**DLM-One-TESS**|**0.2514**|**0.5863**|**0.9380**|**0.93499**|**1**|
>
>     The above results in the table show that when applied to a more powerful teacher model, DLM-One's performance can improve accordingly. DLM-One-TESS significantly accelerates the inference of TESS, reducing the total inference steps from 1000 to 1 while maintaining competitive text generation performance.
>
> 3. **Open-domain Creative Text Generation.**
>
>      Yes, according to our new experiments on unconditional text generation on Plaid, DLM-One can indeed perform such task.
>
> 4. **First One-step Sequence Generation Diffusion Language Model.**
>
>     **Yes, to the best of our knowledge, DLM-One is the first diffusion-based language model that can achieve competitive one-step sequence generation.**
>
> [1] Gulrajani, Ishaan, and Tatsunori B. Hashimoto. "Likelihood-based diffusion language models." *NeurIPS 2023*.
>
> [2] Mahabadi, Rabeeh Karimi, et al. "Tess: Text-to-text self-conditioned simplex diffusion." *EACL 2024*.

---

### Official Review · Reviewer_KzE9 · 2025-10-31

**Soundness:** 2
**Presentation:** 3
**Contribution:** 3
**Rating:** 4
**Confidence:** 2

**Summary:**

This paper proposes a method to distill a diffusion language model for extreme acceleration, enabling generation in just one step, i.e., without iterative refinement. The authors introduce DLM-One, a score-distillation-based framework trained with a two-stage optimization scheme designed to mitigate the latency caused by the alternating update strategy of the score estimator. Experiments are conducted using DiffuSeq as the teacher model and evaluated on three sequence-to-sequence tasks. The results show that DLM-One achieves competitive performance compared to DiffuSeq, with performance gaps ranging from less than 1% to 5%, though it lags in terms of diversity. At the same time, it reduces sampling cost by up to ~500× compared to DiffuSeq.

**Strengths:**

* The paper's focus on achieving one-step sequence generation with diffusion models is well-grounded and addresses a clear need for improved inference efficiency.
* The work demonstrates a strategic adaptation of distillation techniques from vision to language, validating its effectiveness for text generation.
* Through comprehensive ablation studies and discussion, the authors provide convincing verification for their data distillation approach with DiffuSeq.

**Weaknesses:**

* All experiments in the paper were conducted using DiffuSeq. It remains unclear whether the proposed method can generalize effectively to other diffusion-based language models.
* The empirical comparisons are primarily made against the teacher model. It would be valuable to include comparisons with other accelerated generation baselines and provide an analysis discussing the optimal model selection on the performance-efficiency trade-off.

**Questions:**

1. The experiments are condected solely with DiffuSeq. How about the performance with other diffusion language models?
2. Given that several related methods are discussed in Section 2.2, it would be valuable to include a comparative analysis of their performance and inference speed relative to DLM-One, in order to better situate its trade-offs in the landscape of efficient generation models.

---

> ### Author Response · Authors · 2025-11-22
> **Rebuttal (1/2)**
>
> Thank you for your positive comments on DLM-One. We are delighted that you regard the focus of our paper as well-grounded, our adaptation for text generation as effective, and the ablation studies and discussion as comprehensive and convincing. We hope our response can help align your overall positive sentiment with the quantitative evaluation/assessment of DLM-One. Regarding your concerns and questions, we would like to clarify the following:
>
> 1. **Generalization to Other Diffusion-based Language Models.**
>
>    **DLM-One can indeed generalize to other DLMs.** We want to provide experimental results on **Plaid** [1] and **TESS** [2] here as additional pieces of evidence to support DLM-One's generality:
>
>     i. **Plaid** is a state-of-the-art continuous DLM optimized for likelihood metrics (such as NLL). To demonstrate the general applicability of DLM-One beyond DiffuSeq, we have conducted additional experiments using Plaid. Specifically, we pretrained a custom Plaid teacher model on OpenWebText2 and evaluated the GPT-2-based generative perplexity of unconditional generation results using different numbers of inference steps. While the teacher Plaid model can reach a perplexity score of **83.37 using 4096 steps** and **94.28 using 256 steps**, our student DLM-One can achieve a perplexity score of **93.99 using just one step**, which is effectively a **256$\times$** speedup.
>
>     ii. **TESS.** is a recent work from a new family of continuous DLMs (i.e., simplex diffusion), which has shown a competitive performance against other LMs, such as autoregressive models and discrete DLMs. The Tess framework significantly deviates from the conventional modeling practice that optimizes the model in the embedding space. Instead, *TESS operates **in a special input space based on "k-logit simplex" representation,** and it leverages the logits predicted by its backbone transformer model and opt in for **a different type of training loss, i.e., the cross-entropy loss***. However, to demonstrate DLM-One's general applicability, we followed the official configurations to pretrain a TESS model on Paraphrase (PP) task and QQP dataset, and then distilled the pretrained model by adopting the same principle of DLM-One with TESS-specific adaptation. Considering the specialty of TESS and the non-triviality of the adaptation, we will defer the technical details to a separate work currently in progress. Nonetheless, we would like to share some of our preliminary results here:
>
>     |Model|BLEU|Rouge-L|Bert|Dist-1|# Inference Steps|
>     |:--:|:--:|:--:|:--:|:--:|:--:|
>     |DiffuSeq|0.1829|0.5299|0.7932|0.9747|2000|
>     |DLM-One-DiffuSeq|0.1788|0.5265|0.7851|0.9671|1|
>     |**TESS**|**0.3037**|**0.6182**|**0.9496**|**0.9860**|**1000**|
>     |**DLM-One-TESS**|**0.2514**|**0.5863**|**0.9380**|**0.93499**|**1**|
>
>     The above results in the table show that DLM-One can indeed be applied to more complex DLMs with teacher-specific modifications. DLM-One-TESS significantly accelerates the inference of TESS, reducing the total inference steps from 1000 to 1 while maintaining competitive text generation performance. Please note that we didn't perform a thorough parameter tuning for this new experiment, so the current preliminary results on DLM-One-TESS still have lots of room for future improvement.
>
>
> [1] Gulrajani, Ishaan, and Tatsunori B. Hashimoto. "Likelihood-based diffusion language models." *NeurIPS 2023*.
>
> [2] Mahabadi, Rabeeh Karimi, et al. "Tess: Text-to-text self-conditioned simplex diffusion." *EACL 2024*.

---

> ### Author Response · Authors · 2025-11-22
> **Rebuttal (2/2)**
>
> 2. **Compared with Other Models.**
>
>     i. **Comparison with Other Efficient Generation Models.**
>
>     **To the best of our knowledge, DLM-One is the first continuous-diffusion-based language model that can achieve competitive one-step sequence generation.** The focus of paper is to provide a practical way for one-step generation of Gaussian-diffusion-based language models. Furthermore, independent of the actual backbone used, **DLM-One has already achieved the optimal inference cost**, which only consists of 1 inference step. **Therefore, the most relevant performance comparison of DLM-One would be to compare DLM-One to the teacher model and its own variants (e.g., via ablation studies).**
>
>     Regarding the DLM-One variants and the **performance-efficiency trade-off**, we also demonstrated that:
>
>     - Few-step DLM-One at the inference time can further trade generation diversity for quality in a controllable way **in Section 5 "Discussion"** and
>     - Inference-time text augmentation can enhance generation diversity with almost zero computation overhead in **Appendix C.1 "Mitigating Diversity Loss with Text Augmentation."**
>
>     ii. **Comparison with Autoregressive (AR) Baselines.**
>
>     As discussed in **Section 2.2 "Faster Diffusion,"** most of the methods are developed for diffusion models in vision domain. Without innovative strategies like techniques proposed in DLM-One, they are unlikely to work effectively out of box. A more fair comparison would be to compare DLM-One with similar-sized AR models. Below, we provide a detailed qualitative comparison on the text simplification (TS) task with finetuned GPT-2s (also refer to **Table 7 in Appendix D** of our initial submission):
>
>     | Model | BLEU(↑) | R-L(↑) | BERT(↑) | Dist-1(↑) | SelfB(↓) | Div-4(↑) | # Params | Avg. Inf. Time (s) |
>     | :--- | :---: | :---: | :---: | :---: | :---: | :---: | :---: | :---: |
>     | GPT-2 Base FT | 0.3083 | 0.5461 | 0.8021 | 0.9439 | 0.5444 | 0.6047 | 117M | 0.82 |
>     | GPT-2 Large FT | 0.2693 | 0.5111 | 0.7882 | **0.9464** | 0.6042 | 0.5876 | 774M | 2.34 |
>     | DiffuSeq (Teacher) | 0.3622 | **0.5849** | **0.8126** | 0.9264 | **0.4642** | **0.6604** | **91M** | 14.94 |
>     | DLM-One (Student) | **0.3630** | 0.5839 | 0.8084 | 0.9068 | 0.5456 | 0.4098 | **91M** | **0.03** |
>
>     The above results show that DLM-One is not only much faster in sequence generation but can also retain competitive quality compared with AR counterparts.

---

> > ### Comment · Reviewer_KzE9 · 2025-11-28
> >
> > Thank the authors for their detailed response.
> >
> > Given the high perplexity of Plaid, it appears that the current generative quality of diffusion models lags significantly behind state-of-the-art decoder-only models such as Gemma and Qwen.
> >
> > In light of this, my concern is that whether improving the base performance of diffusion models is more crucial than pursuing extreme single-step distillation, especially since the author did not include comparisons with other distillation methods for diffusion models.

---

> > > ### Author Response · Authors · 2025-11-28
> > >
> > > We thank you for your participation in the discussion. We would like to clarify the following points:
> > >
> > > 1. **High perplexity of Plaid.** Please note that the teacher Plaid model we used for our experiment is a **custom 16-transformer-block model**, which is much smaller than the SoTA AR models you mentioned. The purpose of the additional Plaid experiment is to demonstrate the general applicability of DLM-One to different DLMs and other tasks, rather than to compete on perplexity with large-scale models.
> > > 2. **Improving the base performance of DLMs is more crucial.** We clarify that the focus of DLM-One is to **demonstrate the potential for inference acceleration of DLMs, which can be much faster than their AR counterparts** instead of directly improving the teacher DLMs and competing with SoTA AR models such as Gemma and Qwen. As you correctly pointed out, this requires the teacher DLMs to be comparable to these AR models in the first place. This is a separate research line in DLMs that is orthogonal to our work.
> > > 3. **Teacher DLMs lag behind SoTA AR models?** We noticed that while earlier DLMs (including DiffuSeq, Plaid, and TESS) cannot compete with SoTA AR models, a follow-up work, **TESS-2** [3], a large-scale DLM built upon the TESS framework, has shown promising results on standard NLP benchmarks such as **AlpacaEval (63.1)** and **GSM8k (36.5)**. As demonstrated by our experiments on the TESS framework, DLM-One works with this family of models. Thus, our method is ready to accelerate stronger teachers like TESS-2, with the only trade-off being the increased resources naturally required for the larger model size.
> > > 4. **Lack of comparison with other distillation methods.** To the best of our knowledge, DLM-One is the first distillation framework that is successful in **distilling diffusion-based language models for one-step sequence generation**. However, if you are aware of any other works **sharing this same objective** and that are **as effective as DLM-One** for distilling diffusion-based language models, we would be grateful for the references. Please note that the success of distillation methods for general diffusion models (especially in the vision domain) does not directly transfer to the language domain. As mentioned in our response to Reviewer kNgF, distillation in the language domain suffers from two unique challenges: poor initial predictions and variable output length.
> > >
> > > [3] Tae, Jaesung, et al. "Tess 2: A large-scale generalist diffusion language model." *arXiv preprint arXiv:2502.13917* (2025).

---

### Official Review · Reviewer_kNgF · 2025-11-01

**Soundness:** 2
**Presentation:** 2
**Contribution:** 3
**Rating:** 6
**Confidence:** 4

**Summary:**

This paper proposes a practical distillation framework for training continuous diffusion language models for one-step sequence generation (DLM-One), without the need for iterative refinement during generation. The paper conducts three sequence-to-sequence (Seq2Seq) tasks, including question generation (QG), text simplification (TS), and paraphrasing (PP), to support its claims and demonstrate the effectiveness of the framework. However, the novelty of this paper is modest: most components (score distillation, adversarial stabilization, two-stage optimization) are transferred from the vision domain. The experiments are conducted on only three general datasets, which lack a comprehensive study to convince the reader. In addition, the paper is not well organized; for example, mixing the introduction with related work, which makes this paper a bit hard to follow.

**Strengths:**

1. This paper proposes a practical distillation framework for training continuous diffusion language models for one-step sequence generation (DLM-One), without the need for iterative refinement during generation.
2. Three experiments on benchmarks (QQP, Quasar-T, Wiki-Auto) demonstrate competitive BLEU, ROUGE, and BERTScore compared to DiffuSeq, showing the effectiveness and validity of the framework.
3. This method reduces inference cost—up to 500× speedup—without large quality degradation.

**Weaknesses:**

1. Most of the components (score distillation, adversarial stabilization, two-stage optimization) are transferred from the vision domain, which limits the novelty of this paper.
2. All experiments depend on one teacher model (DiffuSeq). The results may not generalize to other DLMs.
3. Lack of experiments on classic generation tasks that require strict semantic evaluation, such as translation.
4. Since degeneration is mentioned, there is limited qualitative or quantitative analysis of when or why the student diverges.
5. In terms of writing, this paper is not well organized, such as mixing related work into the introduction, which makes it hard to follow and somewhat redundant.

**Questions:**

1. Can this framework be used with other teacher models to show generalization?
2. Can more benchmarks be used to explore the generalization of the framework, such as long-content Seq2Seq or translation benchmarks?

---

> ### Author Response · Authors · 2025-11-22
> **Rebuttal (1/2)**
>
> Thank you for your positive feedback. We are encouraged by your comments on the practicality of DLM-One framework for one-step sequence generation and its inference efficiency and competitive performance. Regarding your concerns and questions, we would like to clarify that:
>
> 1. **General Applicability on Other DLMs.** We thank you for raising this question. **DLM-One can indeed generalize to other DLMs.** Apart from DiffuSeq, we further conducted experiments on **Plaid** [1] and **TESS** [2] to demonstrate the effectiveness of DLM-One.
>
>     i. **Plaid** is a state-of-the-art continuous DLM optimized for likelihood metrics (such as NLL). In light of your thoughtful comment, we have conducted additional experiments using Plaid to demonstrate the general applicability of DLM-One beyond DiffuSeq. Specifically, we pretrained a custom Plaid teacher model on OpenWebText2 and evaluated the GPT-2-based generative perplexity of unconditional generation results using different numbers of inference steps. While the teacher Plaid model can reach a perplexity score of **83.37 using 4096 steps** and **94.28 using 256 steps**, our student DLM-One can achieve a perplexity score of **93.99 using just one step**, which is effectively a **256**$\times$ speedup.
>
>     ii. **TESS** is a recent work from a new family of continuous DLMs (i.e., simplex diffusion), which has shown competitive performance against other LMs, such as autoregressive models and discrete DLMs. The Tess framework significantly deviates from the conventional modeling practice that optimizes the model in the embedding space. Instead, *TESS operates **in a special input space based on "k-logit simplex" representation,** and it leverages the logits predicted by its backbone transformer model and utilizes **a different type of training loss, i.e., the cross-entropy loss***. However, to demonstrate DLM-One's general applicability, we followed the official configurations to pretrain a TESS model on Paraphrase (PP) task and QQP dataset, and then distilled the pretrained model by adopting the same principle of DLM-One with TESS-specific adaptation. Considering the specialty of TESS and the non-triviality of the adaptation, we will defer the technical details to a separate work currently in progress. Nonetheless, we would like to share some of our preliminary results here:
>
>     |Model|BLEU|Rouge-L|Bert|Dist-1|# Inference Steps|
>     |:--:|:--:|:--:|:--:|:--:|:--:|
>     |DiffuSeq|0.1829|0.5299|0.7932|0.9747|2000|
>     |DLM-One-DiffuSeq|0.1788|0.5265|0.7851|0.9671|1|
>     |**TESS**|**0.3037**|**0.6182**|**0.9496**|**0.9860**|**1000**|
>     |**DLM-One-TESS**|**0.2514**|**0.5863**|**0.9380**|**0.93499**|**1**|
>
>     The above results in the table show that DLM-One can indeed be applied to more complex DLMs with teacher-specific modifications. DLM-One-TESS significantly accelerates the inference of TESS, reducing the total inference steps from 1000 to 1 while maintaining competitive text generation performance. Please note that we didn't perform a thorough parameter tuning for this new experiment, so the current preliminary results on DLM-One-TESS still have lots of room for future improvement.
>
> 2. **Causes of Student Degeneration.**
>
>     i.  **Poor Initial Predictions.** Unlike in the vision domain, where diffusion model's predictions at a large timestep can be blurry but still recognizable, a DLM’s initial predictions are often incoherent (e.g., gibberish and making little sense), exacerbating the inaccurate score estimation problem of the fake score network and restricting the student's final performance (also mentioned in **Section 3.3 "Two-stage Training"**). Therefore, in DLM-One, we propose *a two-stage training approach* to mitigate this initial score mismatch issue in the second stage.
>
>     ii. **(Root Cause) Variable Output Length.** Unlike multistep models, one-step DLMs must determine the final output length upfront, which makes training unstable and prone to degeneration. One specific issue is related to the use of trailing `[PAD]` tokens. **Without supervision from data, the student generator can easily fail by simply learning to output a target sequence full of `[PAD]` tokens**, because it can trick the teacher model as a "valid" data pattern and lead to very small discrepancy between the teacher model and the fake score model. This issue is also noted in **Section 3.2 "Adversarial Regularization,"** where we introduce *an adversarial loss term* to provide regularization on the generation sequence length.
>
> [1] Gulrajani, Ishaan, and Tatsunori B. Hashimoto. "Likelihood-based diffusion language models." *NeurIPS 2023*.
>
> [2] Mahabadi, Rabeeh Karimi, et al. "Tess: Text-to-text self-conditioned simplex diffusion." *EACL 2024*.

---

> ### Author Response · Authors · 2025-11-22
> **Rebuttal (2/2)**
>
> 3. **Lack of Classic Generation Experiments.** We would like to clarify that the reason we did not include the mentioned generation tasks, such as machine translation (which is also a Seq2Seq task), is that our DLM and AR baselines, including DiffuSeq and Plaid, are not pretrained or finetuned on those specific domains (e.g., WMT 2014 English-to-French data). That being said, we believe our Seq2Seq experiments have already covered the tasks (paraphrase, question generation, and text simplification) that require "strict semantic evaluation." DLM-One has been evaluated on all three Seq2Seq tasks and classic metrics like BLEU, ROUGE-L, and Dist-1.
>
> 4. **Writing of Introduction and Related Work.** We thank the reviewer for the advice. We will revise the "Introduction" section in our new manuscript to avoid redundancy with the actual "Related Work" section.

---

### Official Review · Reviewer_dhub · 2025-11-01

**Soundness:** 2
**Presentation:** 2
**Contribution:** 2
**Rating:** 2
**Confidence:** 4

**Summary:**

This paper introduces a score-based distillation framework, DLM-One, for single-step sequence generation with continuous diffusion language models. By distilling the score function from a pre-trained teacher model into a student model, this approach eliminates the iterative refinement process required by traditional diffusion models, resulting in up to a 5x inference speedup while maintaining generation quality. The authors validate the method's effectiveness on three sequence-to-sequence tasks and demonstrate an approximately 500-fold speedup in text simplification without a significant degradation in quality.

**Strengths:**

1. The proposed method, DLM-One, introduces a novel approach to single-step sequence generation using continuous diffusion language models by leveraging score-based distillation techniques.

2. The authors conduct experiments across three S2S tasks, demonstrating the effectiveness of DLM-One compared to baselines. They also provide thorough analysis of the trade-off between quality and diversity in single-step generation, highlighting potential areas for future improvements.

**Weaknesses:**

1. Lack of Baselines. The paper only discusses one outdated baseline, DiffuSeq (2022), and does not discuss other acceleration techniques.

2. The generation tasks are overly simplistic. The paper does not explicitly discuss the challenges associated with scaling the proposed method to larger datasets or more complex tasks.

**Questions:**

NA.

---

> ### Author Response · Authors · 2025-11-22
>
> We thank you for your feedback on DLM-One. We are glad that you find the DLM-One framework novel and effective for single-step sequence generation and our experiments for Sequence-to-Sequence tasks thorough. Regarding your concerns, we would like to clarify the following:
>
> 1. **Lack of Baselines.**
>
>     i. While there do exist a few recent works on diffusion-based language model, most of them are based on **categorical (discrete) diffusion** [1, 2] or **Riemannian (manifold) flow-matching** [3, 4]. For the most classical, continuous Gaussian-noise based diffusion language models, we believe DiffuSeq is one of the most representative works that embodies the core spirit in using Gaussian diffusion for sequence generation.
>
>     ii. Nevertheless, to address your concern, we further conduct experiments on Plaid [5] on the unconditional generation task (**a more challenging task**) and OpenWebText2 (**a larger scale dataset**) [6]. Specifically, we pretrained a 16-block-transformer Plaid model and distilled it using DLM-One. To evaluate the distillation performance, we compute the generative perplexity using GPT-2.  Please find the experimental results below:
>
>     |# Inference Steps|1 Step|16 Steps|64 Steps|256 Steps|4096 Steps|
>     |:--:|:--:|:--:|:--:|:--:|:--:|
>     |Plaid (Teacher)|-|298.92|122.41|$\underline{94.28}$|**83.37**|
>     |DLM-One-Plaid (Student)|$\underline{93.99}$|-|-|-|-|
>
>     As is shown in the table, the one-step student DLM-One model (denoted as "DLM-One-Plaid") can achieve similar generative perplexity compared to the teacher Plaid model that uses 256 inference steps.
>
> 2. **Discussion of Other Acceleration Techniques.**
>
> 	i. Thank you for your suggestion. We will be sure to discuss those potential acceleration techniques in the revised related work section and clarify their relationship to DLM-One.
>
> 	ii. To the best of our knowledge, although there are acceleration techniques that are used to accelerate diffusion models in vision domain, adapting them effectively to language modeling and one-step sequence generation remains an open challenge. **DLM-One is the first framework to successfully demonstrate the potential of score distillation in continuous Gaussian-diffusion-based language models and one-step sequence generation.**
>
> [1] Sahoo, Subham, et al. "Simple and effective masked diffusion language models." *NeurIPS 2024*.
>
> [2] Lou, Aaron, et al. ‘Discrete Diffusion Modeling by Estimating the Ratios of the Data Distribution’. *ICML 2024*.
>
> [3] Chen, Ricky T. Q., and Yaron Lipman. ‘Flow Matching on General Geometries’. *ICLR 2024*.
>
> [4] Cheng, Chaoran, et al. "Categorical flow matching on statistical manifolds." *NeurIPS 2024*.
>
> [5] Gulrajani, Ishaan, and Tatsunori B. Hashimoto. "Likelihood-based diffusion language models." *NeurIPS 2023*.
>
> [6] Gao, Leo, et al. "The Pile: An 800GB Dataset of Diverse Text for Language Modeling". arXiv preprint arXiv:2101.00027. (2020).

---

### Comment · Area_Chair_fkGK · 2025-11-28

Dear Reviewers,

The authors have responded to your reviews. Please engage in the discussion and evaluate the authors’ rebuttal to check whether your comments have been adequately addressed, and determine whether you would like to adjust your evaluations.

Best,

Your AC

---

### Author Response · Authors · 2025-12-03
**Summary of Rebuttal**

Dear Area Chair,

We would like to provide this summary to assist in your final assessment. The reviewers’ main concerns regarding **DLM-One** focused on generalizability and baselines. In response, we have added new experiments and provided references to the relevant sections of the original submission. We believe that our clarifications and additional results adequately address the weaknesses raised in the initial reviews.

To summarize, we have clarified the following key points of **DLM-One**:

**1. (Reviewers dhub, kNgF, KzE9) General Applicability.** We have demonstrated that DLM-One indeed generalizes to different diffusion-based language models (DLMs) and tasks, including two state-of-the-art continuous DLM frameworks:

- **Plaid:** We distilled a Plaid teacher model on **OpenWebText2** (a larger-scale dataset) for unconditional generation, where our **1-step** student DLM-One model matches the generative perplexity of the **256-step** teacher model, demonstrating massive inference acceleration without clear performance degradation.
- **TESS:** We adapted DLM-One for TESS on the paraphrase task and QQP dataset, where our **1-step** student DLM-One model is competitive with the **1000-step** TESS teacher model (*e.g.,* comparable BLEU, Rouge-L, and BERT score), reducing inference cost by three orders of magnitude.

**2. (Reviewer KzE9) Novelty and Position in Literature.** We clarified that **DLM-One is the first framework to successfully achieve one-step sequence generation for DLMs**. There are no direct existing baselines for this specific task. In addition, we emphasized that **our contribution is the general distillation framework for continuous DLMs** rather than **the direct improvement over the teacher model's base performance**, which is an orthogonal line of research to our work. Our method has successfully accelerated all three teachers (DiffuSeq, Plaid, and TESS) used. Given stronger teacher DLMs (e.g., TESS-2), DLM-One is well positioned to further accelerate them, with the ultimate goal of matching the performance of state-of-the-art autoregressive LLMs, such as Gemma and Qwen, while achieving substantial speedups.

**3. (Reviewer kNgF) Detailed Analysis and In-Context References.** We provided a detailed analysis to dissect the causes for the students' divergence: *poor initial predictions* and *variable output lengths*, and how our *Two-Stage Training* (Section 3.3) and *Adversarial Regularization* (Section 3.2) specifically tackle these issues, respectively.

We have demonstrated that DLM-One is a general framework that unlocks **one-step sequence generation for continuous DLMs**. We achieved up to **~500× speedups** while maintaining teacher quality. We believe these clarifications sufficiently highlight the paper's significance and novelty of efficient sequence generation.

Thank you again for your consideration of our work during this complex review cycle.

Best regards,

The Authors of DLM-One

---

### Meta-Review · Area_Chair_BNLw · 2026-01-07

**Summary:**

The reviewers agree that acceleration of diffusion language models is an interesting and timely problem, and that the proposed framework is technically sound. There is skepticism about the overall significance and importance of the contribution. Questions remain about whether one-step distillation is the right focus given the current quality gap between diffusion LMs and strong autoregressive models. Several reviewers also question whether the paper is properly situated in the context of alternative acceleration or distillation approaches.

**Reviewer Concerns:**

The rebuttal addresses several concerns, including generalization beyond DiffuSeq (Plaid, TESS experiments) and analysis of failure modes such as degeneration and length instability. Higher-level concerns remain unresolved. Reviewers question the strength of the baselines and comparative positioning, limited evaluation in more demanding text generation settings, and whether the proposed contribution is compelling given that continuous diffusion LMs themselves already lag behind state-of-the-art autoregressive models in quality. These issues leave uncertainty about the practical impact of the work.

**Reviewer Scores:**

dhub: likely no change

kNgF: likely remains borderline; balance of concerns is already substantial for 6

KzE9: re-emphasizes concerns in discussion; unlikely to increase

gjET: unlikely to increase to stronger endorsement

---

### Decision · Program_Chairs · 2026-01-26

Reject